# Guanidines: Synthesis of Novel Histamine H_3_R Antagonists with Additional Breast Anticancer Activity and Cholinesterases Inhibitory Effect

**DOI:** 10.3390/ph16050675

**Published:** 2023-04-30

**Authors:** Marek Staszewski, Magdalena Iwan, Tobias Werner, Marek Bajda, Justyna Godyń, Gniewomir Latacz, Agnieszka Korga-Plewko, Joanna Kubik, Natalia Szałaj, Holger Stark, Barbara Malawska, Anna Więckowska, Krzysztof Walczyński

**Affiliations:** 1Department of Synthesis and Technology of Drugs, Medical University of Lodz, Muszyńskiego 1, 90-151 Łódź, Poland; krzysztof.walczynski@umed.lodz.pl; 2Department of Toxicology, Medical University of Lublin, Chodźki 8, 20-093 Lublin, Poland; magdalena.iwan@umlub.pl; 3Institute of Pharmaceutical and Medicinal Chemistry, Heinrich Heine University Düsseldorf, Universitaetsstr. 1, 40225 Duesseldorf, Germany; t.werner@hhu.de (T.W.); stark@hhu.de (H.S.); 4Department of Physicochemical Drug Analysis, Jagiellonian University Medical College, Medyczna 9, 30-688 Kraków, Poland; marek.bajda@uj.edu.pl (M.B.); justyna.godyn@uj.edu.pl (J.G.); natalia.guzior@uj.edu.pl (N.S.); mfmalaws@cyf-kr.edu.pl (B.M.); anna.wieckowska@uj.edu.pl (A.W.); 5Department of Technology and Biotechnology of Drugs, Jagiellonian University Medical College, Medyczna 9, 30-688 Kraków, Poland; gniewomir.latacz@uj.edu.pl; 6Independent Medical Biology Unit, Medical University of Lublin, Jaczewskiego 8b, 20-090 Lublin, Poland; agnieszka.korga-plewko@umlub.pl (A.K.-P.); joanna.kubik@umlub.pl (J.K.)

**Keywords:** guanidines, histamine H_3_ receptor, antagonist, breast cancer, multi-target directed ligand

## Abstract

This study examines the properties of novel guanidines, designed and synthesized as histamine H_3_R antagonists/inverse agonists with additional pharmacological targets. We evaluated their potential against two targets viz., inhibition of MDA-MB-231, and MCF-7 breast cancer cells viability and inhibition of AChE/BuChE. ADS10310 showed micromolar cytotoxicity against breast cancer cells, combined with nanomolar affinity at *h*H_3_R, and may represent a promising target for the development of an alternative method of cancer therapy. Some of the newly synthesized compounds showed moderate inhibition of BuChE in the single-digit micromolar concentration ranges. H_3_R antagonist with additional AChE/BuChE inhibitory effect might improve cognitive functions in Alzheimer’s disease. For ADS10310, several in vitro ADME-Tox parameters were evaluated and indicated that it is a metabolically stable compound with weak hepatotoxic activity and can be accepted for further studies.

## 1. Introduction

In recent years, multi-target directed ligands (MTDLs) have become a significant target of research into treatments for diseases with complex pathogenesis such as cancer or Alzheimer’s disease (AD). MTDLs are compounds that are able to simultaneously modulate more than one target [1]. These compounds may be more effective and less vulnerable to adaptive resistance and may also offer lower toxicity resulting from fewer drug–drug interactions (DDIs) and a unified pharmacokinetic profile. In addition to mediating the inhibition of synthesis and release of histamine from histaminergic neurons via a negative feedback loop, the histamine H_3_ receptors (H_3_R), i.e., presynaptic heteroreceptors in the brain, are involved in the regulation of numerous other neurotransmitter systems, including acetylcholine, dopamine, norepinephrine, serotonin, γ-aminobutyric acid, glutamate, and substance P [2,3].

The growing number of cancer patients, and the ineffectiveness of therapies, has driven the search for new and more effective treatments, including those acting on targets previously not considered anti-cancer. The involvement of histamine and histamine receptors in cancer cell proliferation has also been reported [4,5,6,7,8,9,10]. The effect of histamine on tumor growth depends on the expression of the functional histamine receptors directly on the surface of tumor cells or in cells surrounding the tumor [11]. H_3_R expression has been found in several cancer cell lines including glioblastoma (GBM), [12] McA-RH7777 hepatoma cells, [13] and MDA-MB-231 and MCF-7 breast cancer cells [14]. It was found that H_3_R mRNA and protein levels were up-regulated in the GBM and glioma cell lines compared to normal brain tissue and astrocytes [12]. Studies have also examined the effect of H_3_R antagonists/inverse agonists including ciproxifan, pitolisant (Figure 1), clobenpropit, thioperamide, FUB349, and FUB465 on McA-RH7777 hepatoma cell proliferation [13]. The role of H_3_R in human mammary carcinogenesis and the importance of histamine in breast cancer have still been under discussion [15]. Histamine has also been implicated in the modification of the invasive phenotype in MDA-MB-231 breast cancer cells by decreasing cell adhesion and altering the balance between matrix metalloproteinase 9 (MMP-9) and tissue inhibitor of metalloproteinase 2 (TIMP-2) [16,17]. Immunohistochemical analysis confirmed H_3_R expression in 67% of benign lesions and 95% of studies of human mammary gland epithelium carcinomas [15]. The intracellular content of histamine in human breast cancer was found to be higher in MDA-MB-231 cells than in MCF-7 cells [18]. It has been found that histamine modulates the proliferation of MDA-MB-231 breast cancer cells [18]. At 10 µM concentration, histamine remarkably suppresses the proliferation of MDA-MB-231 breast cancer cells and enhances radiosensitivity, whereas 0.01 µM moderately increases cell proliferation [18]. Some H_3_R antagonists also modulate breast cancer proliferation; for example, JNJ5207852 (Figure 1) and thioperamide inhibit histamine-induced breast cancer cell proliferation. Hence, H_3_R represents a promising target for the development of an alternative method of cancer therapy.

Stimulation of histamine H_3_ heteroreceptors in the central nervous system (CNS) by H_3_R agonists (imetit, immepip) diminishes acetylcholine (ACh) release, while blocking H_3_ autoreceptor activity with the H_3_R antagonist (thioperamide) increases ACh release [19]. Several H_3_R antagonists with acetylcholinesterase/butyrylcholinesterase (AChE/BuChE)-inhibitory activity have been reported (Cpd. 13 [20], Cpd. 41 [21]—Figure 1), and might serve as novel important tools for further pharmacological investigations on histaminergic neurotransmission and its regulatory processes [22]. An MTDL designed by the fusion of an H_3_R antagonist and an AChE/BuChE inhibitor in a single molecule might improve cognitive functions in AD [23].

## 2. Results and Discussion

The aim of the present study was to design novel guanidine-based H_3_R antagonists/inverse agonists acting on additional biological targets. One of the additional evaluated targets was inhibiting nonestrogen-responsive MDA-MB-231, or estrogen-responsive MCF-7 breast cancer cell viability. The second investigated target was inhibiting AChE or BuChE, which might improve cognitive functions in AD.

### 2.1. Design

Based on our previously obtained in silico studies for ADS1017 and ADS1020 we have identified that the benzylguanidine fragment demonstrates a universal match for the recognition site of specific H_3_R ligands [24]. Therefore, the phenoxyalkyl substituent of ADS1017 and ADS1020 was modified by introducing a chlorine atom into the 4-position in the aromatic ring, and replacing the aryloxyheptyl substituent with an arylpropyloxypropyl, similarly as in the structure of pitolisant (Figure 1). Another modification of newly synthesized compounds are derivatives in which the oxygen of the aliphatic linker of pitolisant has been replaced with a guanidine fragment, substituted in the 1,1 or 1,3 positions. The length of the alkyl chains between the central guanidine molecule, and piperidine or 4-chlorophenyl ring was elongated by three to four methylene groups (Figure 1). This series is also a modification of OUP-186 (Figure 1), where the isothiourea fragment is the central part of the molecule [25]. OUP-186 has been described as a potent H_3_R antagonist, but also a compound that effectively reduces the proliferation of MCF-7, and MDA-MB-231 breast cancer cells [14].

### 2.2. Chemistry

As illustrated in Figure 2, etherification of commercially available 4-chlorophenol and 1,7-dibromoheptane with sodium phenoxide in anhydrous ethanol led to compound **1**. Compound **2** was obtained from **1** by alkylation with piperazine. *N*-alkylation of **2** with 4-bromobutyronitrile in the presence of potassium carbonate in acetonitrile led to the formation of **3**. The synthesis of compounds illustrated in Figure 3 began by bromination of commercially available 3-phenyl-1-propanol with phosphorus tribromide in toluene, and substitution 3-(4-chlorophenyl)-1-propanol under the condition of triethylamine in DCM with methanesulfonyl chloride, leading to **8** or **9** semi-products, respectively. Alkylation of piperazine with 4-bromobutyronitrile, followed by reaction with commercially available 3-bromo-1-propanol afforded compound **11**. Etherification of **11** with **8** or **9** led to the appropriate nitrile derivatives **12a**–**b**.

Further synthetic procedures were similar for all compounds presented in Figure 2 and Figure 3. Nitriles (**3**, **12a**–**b**) were reduced with LiAlH_4_ in dry diethyl ether to corresponding primary amines (**4**, **13a**–**b**). *N*-acylation with commercially available benzoyl chloride or 4-(trifluoromethyl)benzoyl chloride in the presence of triethylamine led to the synthesis of amides (**5a**–**b**, **14a**–**d**), which were subsequently reduced with LiAlH_4_ to the secondary amines (**6a**–**b**, **15a**–**d)**. Guanylation of secondary amines (**6a**–**b**, **15a**–**d**) with 1,3-bis(*tert*-butoxycarbonyl)-2-methylisothiourea in the presence of triethylamine and 10% excess of mercury (II) chloride resulted in **7a**–**b** and **16a**–**d**. The final compounds were obtained by acidic deprotection of Boc-groups from guanidine moiety, resulting in ADS10377, ADS10376, ADS10349, ADS10350, ADS10278, and ADS10279.

Intermediate primary amines (**18a–b**) and bromides (**19**, **21**) (Figure 4) was obtained first to synthesize compounds presented in Figure 5 and Figure 6. Alkylation of piperidine with acrylonitrile in methanol or 4-bromobutyronitrile in the presence of potassium carbonate in acetonitrile led to the formation of **17a**–**b**, respectively. Both nitriles were reduced with LiAlH_4_ to primary amines **18a**–**b**. Reduction of commercially available 3-(4-chlorobenzoyl)propionic acid with borane-*tert*-butylamine-complex in the presence of aluminum chloride in DCM led to obtaining **20** [26]. Bromination of 3-(4-chlorophenyl)-1-propanol and **20** with phosphorus tribromide in the toluene led to the formation of intermediate bromides **19** and **21**, respectively.

*N*-Alkylation of primary amines (**18a**–**b**) with appropriate bromides (**19, 21**) in the presence of potassium carbonate in acetonitrile led to the synthesis of the secondary amines **22a**–**c**. Guanylation with 1,3-bis(*tert*-butoxycarbonyl)-2-methylisothiourea in the presence of triethylamine and 10% excess of mercury (II) chloride resulted in the deprotection of Boc-groups from guanidine moiety, resulting in the synthesis of ADS10292, ADS10300, ADS10312 (Figure 5). Treatment of 3-(4-Chlorophenyl)-1-propanol or **20** with 1,3-bis(*tert*-butoxycarbonyl)-2-methylisothiourea and 94% diisopropyl azodicarboxylate (DIAD) in the presence of triphenylphosphine in THF resulted in **24a**–**b**, respectively [27]. They were then reacted with **18a**–**b** in a mixture of THF–water to give compounds **25a**–**d**. Finally, compounds **25a**–**d** were treated with HCl in dioxane to cleave the Boc moieties, affording the final compounds ADS10298, ADS10301, ADS10306, and ADS10310 (Figure 6). 

### 2.3. Pharmacology

#### 2.3.1. Ex Vivo Screening of Histamine gpH_3_R/gpH_1_R Antagonists/Inverse Agonists on Guinea Pig Ileum and hH_3_R Radioligand Displacement Assay

All newly synthesized compounds were evaluated as H_3_R antagonists/inverse agonists on the guinea pig ileum (*gp*H_3_R), stimulated electrically to induce contractions. The distinct affinity profile and species-dependent pharmacology of H_3_R ligands are known [28]. Furthermore, OUP-186 showed high affinity to *h*H_3_R and no histamine release in the rat brain in an in vivo microdialysis experiment [25]. In contrast, ADS003 (Figure 1) demonstrated higher affinity to rat *r*H_3_R than to human *h*H_3_R [29]. Therefore, to compare species differences, all newly synthesized compounds were subjected to radioligand displacement assay in membrane fractions of HEK-293 cells stably expressing *h*H_3_R. Results obtained by the *h*H_3_R displacement binding assay and on the isolated guinea pig ileum were generally comparable according to structure-activity relationship (SAR). Compounds based on Formula **B** were less potent than those based on Formula **A** (Table 1). In contrast to results obtained on the guinea pig ileum, the presence of chlorine gave compounds a slightly higher affinity to the *h*H_3_R. Based on general Formulas **C** and **D**, a higher affinity was observed for 1,3-disubstituted guanidines. The length of the alkyl chain also influences pA_2_ value. Compounds with four methylene groups between the guanidine substituent and piperidine ring are more active than those with three methylene groups. *h*H_3_R radioligand displacement assay confirmed the results of the isolated guinea pig ileum. Three of the tested compounds (ADS10301, ADS10306, ADS10310) showed *K_i_* values below 200 nM. The most active compound was ADS10310 (*gp*H_3_R pA_2_ = 7.72; *h*H_3_R *Ki* = 127 nM). 

A second histamine receptor located in the guinea pig ileum is H_1_R. The antagonistic effect at H_1_R was measured on the isolated guinea pig ileum stimulated to contractions by histamine [30]. To eliminate the impact of muscarinic receptors located on the tested tissue, 0.05 µM of atropine was added to the Krebs buffer. All compounds showed low to moderate affinities at H_1_R. Only for ADS10306 was the pA_2_ value over 7, but it still demonstrated only moderate affinity. Since no high potency was observed, no further investigations on the H_1_R were undertaken.

#### 2.3.2. Cytotoxicity Analyses

The selected compounds (ADS1017, ADS10310) were qualified in MCF-7, and MDA-MB-231 breast cancer cell viability assays. The cytotoxicity of the tested compounds against two breast cancer cell lines and normal fibroblasts (BJ) was evaluated by MTT assay. After 48 h of incubation, both compounds were shown to be cytotoxic to both breast cancer cells, with ADS1017 showing a much stronger effect than ADS10310 (Figure 2, Table 1). Based on the comparison of IC_50_ values against BJ and cancer cell lines (MDA-MC-231 or MCF-7), selectivity indexes (SI) were calculated, with SI values greater than 1 indicating that the tested compound is more cytotoxic to the cancer line than to the normal, reference, cell line. In this case, both tested compounds were found to be toxic to normal cells. However, a selectivity index was over 1. For ADS1017 the SI values were 1.96 for MDA-MB-231 and 1.58 for MCF-7. Selectivity indexes were more beneficial for ADS10310, and in both cases exceeded a value of 2 (2.01 and 2.79, respectively). The tested compounds are active at much higher concentrations than the classical chemotherapeutic agent Doxorubicin, but its selectivity index values are relatively low (2.63 and 2.20, respectively) and close to those obtained for ADS10310 (Appendix A).

The MTT test results were verified by microscopic observations of cell morphology after 48 h of treatment with the tested compounds at concentrations corresponding to IC_50_ value. The control cultures of both cell lines revealed a normal epithelial-like morphology specific to each particular cell line and were well adherent. Cells treated with both tested compounds became round, shrank, and floated in the medium, which is a manifestation of cell death (Figure 3a).

#### 2.3.3. Apoptosis Detection

The MTT test results prompted further study. Staining with propidium iodide and Annexin V-FITC was performed to distinguish apoptotic from necrotic cell death. Image cytometry showed that the tested compounds exclusively induce apoptosis in the breast cancer cells. After ADS10310 treatment, mainly late apoptotic cells were observed in both breast cancer lines. In the case of ADS1017 treatment, more apoptotic cells were observed in MDA-MB-231 (39% of the late apoptotic cells population and 22% of the early apoptotic cells) than in MCF-7 (38% of the late apoptotic cells population and 2% of the early apoptotic cells) cell cultures (Figure 3b).

#### 2.3.4. Cell Cycle Analysis

Cell cycle analysis showed that none of the tested compounds induced cell cycle inhibition. However, an increase was observed in the subG1 population, which corresponds to dead cells. The largest peak for the subG1 population was observed after ADS1017 treatment of MDA-MB 231 cells that correspond to the apoptosis detection results (Figure 3c). 

In conclusion, both compounds revealed cytotoxicity against MDA-MB-231, and MCF-7—HER2-negative cells, differing in the expression of estrogen and progesterone receptors. ADS1017 showed stronger activity in the MTT assay, which was confirmed by apoptosis/necrosis detection and cell cycle analysis. However, ADS10310 was relatively less toxic for normal cells.

#### 2.3.5. Inhibition of Electric Eel AChE and Equine Serum BuChE

As synthesized H_3_R antagonists remain within the scope of interest as an MTDL for the treatment of cognition deficit disorders, all compounds were subjected to AChE, and BuChE enzyme inhibition assays. The influence of all compounds on the inhibition of electric eel AChE and equine serum BuChE was evaluated in the spectroscopic Ellman’s assay, modified for 96-well microplates [31]. After prescreening at a concentration of 10 µM, for all compounds with at least 50% inhibition of the enzyme activity, the IC_50_ values were determined. Only one compound (ADS10292) showed over 50% *ee*AChE inhibition (IC_50_ value = 10.88 µM), however eleven compounds were qualified for determination of the IC_50_ value at *eq*BuChE. Comparing Formulas **A** and **B**, it was observed that the incorporation of a chlorine atom at position 4 of the aromatic ring increases the inhibitory effect against *eq*BuChE. IC_50_ values for compounds based on Formula **C** were between 2.4–8.4 µM. A higher inhibitory effect was observed for compounds based on Formula **D** (values: 1.6–5.9 µM) (Table 1). It is worth noting that compounds belonging to this group showed nanomolar potency at *h*H_3_R. This gives a good base for optimizing the structure for further investigation of MTDLs that may be useful in the treatment of e.g., Alzheimer’s disease.

### 2.4. In Silico Studies

Molecular docking studies were performed to characterize the binding mode for the presented compounds and to identify the essential structural elements for the interaction with the previously published homologous model of the H_3_Rs [32]. The analyzed compounds presented very consistent binding modes in the groups of piperidine and piperazine derivatives of disubstituted guanidines. All of them interacted both within orthosteric and allosteric binding sites. Furthermore, they created crucial contacts with amino acid residues which recognize the histamine molecule i.e., glutamate E206 (E5.46) which binds the imidazole ring and aspartate D114 (D3.32) which binds the amine group from histamine. For ADS10310 and the other compounds based on Formulas **C** and **D**, the most important features are the salt bridge between piperidine and glutamate E206 (E5.46), the salt bridge between guanidine and aspartate D114 (D3.32), and aromatic ring interactions of phenyl moiety with tyrosines Y189, Y194, and phenylalanine F193 (Figure 4). Moreover, cation-π interactions were found between guanidine and tryptophan W110 (W3.28) and tyrosine Y115 (Y3.33). In the case of the compounds based on Formulas **A** and **B**, the same amino acid residues were involved in the binding. However, the spatial organization of compound molecules was different, and the guanidine fragment interacted with glutamate E206 (E5.46), while piperazine was engaged in a salt bridge with aspartate D114 (D3.32). The phenyl moiety, similar to the compounds based on Formulas **C** and **D**, interacted with aromatic residues from the extracellular loop ECL2. Moreover, guanidine formed cation−π contacts with tyrosine Y167 (Y4.57) and a hydrogen bond with threonine T119 (T3.37). The benzyl fragment occupied a hydrophobic pocket built by F193, L199 (5.39), W371 (6.48), and M378 (6.55). A similar binding mode was described previously for the compounds based on Formulas **A** and **B** (e.g., compound ADS1017) [24]. For all compounds, we found that the mode of guanidine substitution (1,1 or 1,3) as well as the length of alkyl linkers influenced the quality of crucial interactions. The 1,3-disubstituted guanidine derivatives from the compounds based on Formulas **C** and **D** were generally better adapted in the binding pocket. Furthermore, the long and flexible aliphatic linkers in ADS1017 allowed the phenoxy group to create interactions with the amino acids of the allosteric site such as Y189 (Y45.51) (π–π) and R381 (R6.58) (cation−π). Moreover, the introduction of some substituents, such as a chlorine atom in the phenyl moiety or a trifluoromethyl group in the benzyl fragment, led to extra contacts which might improve the activity.

### 2.5. In Vitro Metabolic Stability

In addition to biological activity, drug-likeness is equally important in drug discovery, and should be considered at the early stage of the drug development process. Therefore, ADS10310 was subjected to testing for several ADME-Tox parameters, including metabolic stability. Compound ADS10310 was found to be metabolically stable compared to the reference unstable drug Verapamil after 120 min of incubation with human liver microsomes (HLMs). The percentage of ADS10310 remaining in the reaction mixture was much higher than that of Verapamil (73.6% and 30.8%, respectively) (Table 2, Figure 5). Moreover, ADS10310 was found to be metabolized into three metabolites (Figure 5). MS spectra indicated the most probable metabolic pathways, including decomposition and triple hydroxylation (main metabolite M1), hydroxylation (M2), and dehydrogenation (M3) (Table 2). The fragmentation of substrate and metabolites occurring during MS analyses also allowed the determination of the most probable parts of ADS10310 hydroxylations and dehydrogenations (Appendix A). These results were also confirmed in silico by MetaSite 6.0.1. software, which predicted the most probable sites of ADS10310 metabolism (Appendix A Appendix A).

### 2.6. The Influence on CYP3A4 and CYP2D6 Activity

Treatment of elderly patients who suffer from cancer or degenerative CNS diseases usually requires the use of more than one medication at the same time, which can result in a greater risk of DDIs. Hence the effect of ADS10310 was measured on two cytochrome P450 (CYP) isoforms, 3A4 and 2D6, the most involved in drug metabolism. The results were compared to the selective CYP3A4 inhibitor ketoconazole and selective CYP2D6 inhibitor quinidine. ADS10310 did not show any influence on CYP3A4 activity (Figure 6a), but strongly inhibited CYP2D6, decreasing its activity by more than 50% even at the lowest tested dose of 0.1 µM (Figure 6b). The data indicate a high risk of potential DDIs after co-administration of ADS10310 with drugs metabolized by CYP2D6.

### 2.7. Hepatotoxicity Assay

The hepatotoxicity of ADS10310 was tested against the hepatoma HepG2 cell line. A statistically significant decrease in cell viability was observed after incubation of ADS10310 for 72 h, but only at the highest dose of 100 µM (Figure 6c). In general, this test showed weak hepatotoxic activity of ADS 10310.

## 3. Materials and Methods

### 3.1. Chemistry

All solvents were purchased from commercial suppliers (e.g., Avantor Performance Materials Poland S.A., PPH Stanlab Sp. z o.o. Lublin, Chempur Piekary Slaskie) and were used without further purification. The methanesulfonyl chloride (Aldrich), 3-phenyl-1-propanol (Fluorochem), 3-bromo-1-propanol (Fluorochem), 3-(4-chlorophenyl)-1-propanol (Fluorochem), 3-(4-chlorobenzoyl)propionic acid (Aldrich), phosphorus tribromide (Aldrich), piperazine (Alfa Aesar), piperidine (Aldrich), 4-bromobutyronitrile (Fluorochem), acrylonitrile (Aldrich), tetra-n-butylammonium iodide (TBAI) (Aldrich), borane-*tert*-butylamine-complex (Acros Organics), lithium aluminum hydride (Aldrich), Sodium hydride (60% dispersion in mineral oil) (TCI), 4-(trifluoromethyl)benzoyl chloride (TCI), benzoyl chloride (Aldrich), 1,3-bis(*tert*-butoxycarbonyl)-2-methylisothiourea (Fluorochem), 4M solution HCl in dioxane (Fluorochem), acrylonitrile (Aldrich), triphenylphosphine (Alfa Aesar), 94% diisopropyl azodicarboxylate—DIAD (TCI)*,* 4-chlorophenol (Aldrich), and 1,7-dibromoheptane (TCI) were purchased from commercial suppliers and used without further purification. Nuclear magnetic resonance (NMR) spectra (^1^H NMR, ^13^H NMR) were recorded on a Bruker Avance III 600 MHz (^1^H NMR spectra were run at 600 MHz, while ^13^C NMR spectra were run at 150.95 MHz) spectrometer in CDCl_3_, and CD_3_OD. Chemical shifts were expressed in δ values—parts per million (ppm) using the solvent signal as an internal standard and coupling constants (*J*) were given in hertz (Hz). Spectra obtained in deuterated chloroform were referenced to tetramethylsilane at 0.00 ppm for ^1^H spectra and 77.02 ppm for ^13^C spectra. Spectra obtained in CD_3_OD were referenced to residual CD_3_OD at 4.84 ppm (methanol-d4; ^1^H singlet) ppm for ^1^H spectra and 49.05 ppm for ^13^C spectra. Signal multiplicities were characterized as: br (broad), s (singlet), d (doublet), t (triplet), q (quartet), qt (quintet), m (multiplet), and * (exchangeable by deuterium oxide). Elemental analysis (C, H, and N) for all compounds were measured on a Perkin Elmer Series II CHNS/O Analyzer 2400 and were within ±0.4% of the theoretical values. Reactions were monitored by thin-layer chromatography (TLC) on silica gel 60 F254 plates (Merck) and visualized using a UV Lamp (254 nm) and cerium molybdate stain. Flash-column chromatography was performed using silica gel 60 Å 50 mm (J. T. Baker B. V.) and Normasil 60 Silica gel 40–63 µm (VWR Chemicals), employing eluent indicated by TLC. Melting points (mp) were measured in open capillaries on an Electrothermal apparatus (Electrothermal, Southend, England) and are uncorrected.

Details of the ADS compounds and semi-product synthesis including NMR spectra are presented in Appendix A. The synthesis of ADS1017 and ADS1020 were synthesized according to Staszewski et al. [34].

### 3.2. Biological Activity

Ex vivo assay for histamine H_3_R and H_1_R antagonists were measured on the isolated guinea pig ileum stimulated to contractions by histamine [30]. The pA_2_-values were calculated according to Arunlakshana and Schild [35]. The radioligand-displacement binding assay was performed in membrane fractions of HEK-293 cells stably expressing hH_3_R. Cell cultivation and membrane preparation was performed according to Kottke et al. [36]. The Ki values were calculated from the IC_50_ values using the Cheng–Prusoff equation [37]. The statistical calculations were performed on –log(K_i_). The mean values and 95% confidence intervals were transformed to nanomolar concentrations. The inhibitory activities of novel ADS compounds were assessed in spectrophotometric Ellman’s assay using AChE from electric eel (eeAChE) and BuChE from horse serum (eqBuChE) [31].

#### 3.2.1. Cell Viability

##### Cell Culturing

The studies were conducted on two breast cancer cell lines—MCF-7 (ER+PR+/HER2-negative) and MDA-MB-231 (ER/PR/HER2-negative) and human normal fibroblast cell line BJ. The cell lines were obtained from American Type Culture Collection (ATCC, Manassas, VA, USA) and were cultured at 37 °C in the presence of 5% CO_2_ in the atmosphere. The culture media (Eagle’s Minimum Essential Medium for MCF-7 and BJ, Leibovitz’s L-15 Medium for MDA-MB-231, Corning, NY, USA), was supplemented with 10% fetal bovine serum. The cell’s morphology was examined under a Nikon Eclipse Ti inverted phase contrast microscope (Nikon, Tokyo, Japan). The authenticity of tested cell lines was verified by short tandem repeat (STR) genotyping in the Department of Forensic Medicine (Medical University of Lublin, Lublin, Poland).

##### Cytotoxicity Evaluation—MTT Test

Cell viability based on metabolic activity was evaluated with the MTT colorimetric test based on the ability of viable cells to cause the transformation of tetrazolium salts MTT (3-[4,5-dimethylthiazol-2-yl]-2,5)diphenyltetrazolium bromide) into purple formazan, by cellular dehydrogenases. The cells were seeded into 96-well plates at a concentration of 2 × 10^5^ cells/mL. Once 70–80% confluency was reached, tested compounds were added: ADS10310 and ADS1017 in concentrations ranging from 1 to 250 µM or DMSO as the vehicle in control cultures for the next 48 h (max. DMSO concentration <0.5%). Doxorubicin was used for comparison as the standard chemotherapeutic agent used in breast cancer treatment. After 48 h of incubation cell cultures were supplemented with 20 µL of 5 mg/mL MTT (Thermofisher, Waltham, MA, USA) stock in PBS (Corning, NY, USA), and the incubation was continued for 4 h at 37 °C. Next, the medium with MTT was removed, and obtained crystals were dissolved in 200 µL of DMSO (dimethyl sulfoxide, Avantor, Gliwice, Poland). The solution absorbency was measured at 570 nm, using a PowerWave™ xs microplate spectrophotometer (BioTek Instruments, Winooski, VT, USA). The experiment was performed three times with three replicates for each concentration of tested compounds. IC_50_ values were determined using the AAT Bioquest IC_50_ calculator (AAT Bioquest, Inc. Quest Graph™ EC50 Calculator. https://www.aatbio.com/tools/ec50-calculator) (accessed on 6 September 2022).

##### Cell Cycle Analysis

Cell cycle analysis was performed using the NucleoCounter^®^ NC-3000™ (Chemometec Inc., Lillerød, Denmark) fluorescence image cytometer. Cells were seeded into 6-well plates (2 × 10^5^ cells/mL) and cultured until 70–80% confluence was reached. Then the cells were incubated with tested compounds in concentrations corresponding to the IC_50_ values obtained in the MTT test or DMSO as the vehicle in control cultures and analyzed after 48 h of incubation using NC 3000 system according to the manufacturer’s protocol for the two-step cell cycle analysis (ChemoMetec Inc., Lillerød, Denmark). Data were derived from three independent experiments with three replicates.

##### Cell Apoptosis Assay

Apoptosis was analyzed using the NucleoCounter^®^ NC-3000™ (Chemometec Inc., Lillerød, Denmark) according to the manufacturer’s protocol using Annexin V from Biotium (Fremont, CA, USA). MCF-7 and MDA-MB-231 cells were seeded into 6-well plates (2 × 10^5^ cells/mL) and cultured until 70–80% confluence was reached. Then the cells were incubated with tested compounds in concentrations corresponding to the IC_50_ values obtained in the MTT test or DMSO as the vehicle in control cultures and analyzed after 48 h incubation. Scatter plots were used to determine the proportion of healthy cells (Annexin V-negative/PI-negative cells), early apoptotic cells (Annexin V-positive/PI-negative cells), late apoptotic cells (Annexin V-positive/PI-positive cells), and necrotic cells (Annexin V-negative/PI-positive cells). Data were derived from three independent experiments with three replicates.

##### Statistical Analysis

Statistical analysis was performed using STATISTICA 13 software (StatSoft, Kraków, Poland). The data were calculated as mean ± SD. Student’s t-test was used for the comparison of MTT test results between control and treated cells. A *p*-value of less than 0.05 was considered statistically significant.

### 3.3. In Silico Studies

The three-dimensional structures for ligands were prepared with LigPrep (Schrodinger Suite) from SMILES strings, using an ionizer for the prediction of protonation states at pH 7.4 ± 0.2. The previously prepared homology model of the histamine H_3_ receptor was utilized for docking studies [32]. This model was built with Modeller 9.14 based on a template of M_3_ muscarinic receptor (PDB code: 4U15) using automodel class and very fast refinement option. Recently, the crystal structure of H_3_R was obtained [38]. Comparing this experimental structure with our homology model, it is worth noting that the model is very consistent with the X-ray structure. Therefore, the application of our homology model in presented docking studies was fully justified. All dockings were performed using the Induced Fit Docking procedure (Schrodinger Suite) with the standard protocol. Binding site box with a center at Asp3.32 and size, adjusted for docking of ligands with length ≤25 Å, was applied. The obtained ligand–receptor complexes were analyzed concerning IFDScore as well as interaction networks with Maestro (Schrodinger Suite) and PyMOL 0.99rc6 (DeLano Scientific LLC).

### 3.4. In Vitro Metabolic Stability

All assays and protocols used for the determination of compound ADME-Tox parameters were described previously [33,39]. In brief, the metabolic stability assay was performed on HLMs, purchased from Sigma-Aldrich (St. Louis, MO, USA). ADS10310 was incubated with HLMs for 120 min in Tris buffer (pH = 7.4) at 37 °C. NADPH regeneration system Promega (Madison, WI, USA) was added to initiate whereas methanol was added to terminate the reaction. The LC/MS analyses of the centrifuged reaction mixture were obtained on Waters ACQUITY™ TQD system (Waters, Milford, CT, USA). The in silico prediction of most probable sites of ADS10310 metabolism was performed by MetaSite 6.0.1 software (Molecular Discovery Ltd., Hertfordshire, UK). To predict potential DDIs the CYP3A4 and CYP2D6 P450-Glo™ kits provided by Promega (Madison, WI, USA) were used according to provided protocols. The results were calculated from two independent experiments.

Cell-based safety tests were performed with hepatoma HepG2 (HB-8065™) cell line obtained directly from ATCC^®^ (American Type Culture Collection, Manassas, VA, USA). The cells were grown in Minimum Essential Media (MEM) supplemented with 10% fetal bovine serum (FBS) in a humidified atmosphere of 5% CO_2_ to reach 70–80% confluency. The cells were seeded next into 96-well plates at a concentration of 0.7 × 10^4^ cells/well. The CellTiter 96^®^ AQueous Non-Radioactive Cell Proliferation Assay (MTS, Promega, Madison, WI, USA) was used for determination of cells viability after 72h of incubation with ADS10310 in concentration range 0.01–100 µM or the reference drug Doxorubicin (DOX, 1 µM). The 1% DMSO in culture media was used as the vehicle control. The results were calculated from two independent experiments. GraphPad Prism 8.0.1 was used for calculation of the IC_50_ value.

The luminescent signal and the absorbances (measured at 490 nm) in DDIs and safety assays were measured by using an EnSpire PerkinElmer microplate reader (Waltham, MA, USA). All reference drugs used—ketoconazole, quinidine, doxorubicin, and verapamil—were purchased from Sigma-Aldrich (St. Louis, MO, USA).

All graphs of the *gp*H_3_R, *gp*H_1_R ex vivo assays, *h*H_3_R radioligand displacement assays, and inhibition of AChE/BuChE are presented in Appendix A. The cell viability scores (%) of MDA-MB-231 and MCF-7 breast cancer cells after ADS10310 and ADS1017 treatment are presented in Appendix A. UPLC spectra of the control reaction, and MS spectra of ADS10310 and the most probable structure of metabolites (in vitro metabolic stability assay) are presented in Appendix A.

## 4. Conclusions

ADS1017 and ADS10310 showed micromolar cytotoxicity against MDA-MB-231, and MCF-7 breast cancer cell lines, combined with nanomolar potency at *h*H_3_R. ADS1017 showed higher toxicity in the MTT assay; however, ADS10310 was relatively less toxic for normal and hepatoma HepG2 cells. Image cytometry analysis showed that the tested compounds exclusively induce apoptosis in breast cancer cells. ADS10310 showed similar selectivity to doxorubicin. Doxorubicin is active in lower concentrations, but the tested compounds should be considered as having dual activity: H_3_R antagonists/inverse agonists, and cytotoxicity against MDA-MB-231, and MCF-7 (breast cancer) cells. These compounds are promising for further optimization and biological profiling. Modifications of ADS1017, and ADS10310 should focus on improving the selectivity index value to reduce cytotoxicity against non-cancerous fibroblasts. Histamine and H_3_R play a role in the breast cancer cells proliferation and H_3_R ligands may represent a promising target for the development of an alternative method of cancer therapy.

The inhibitory activity of the novel compounds was assessed by AChE/BuChE spectrophotometric assay. Only one compound (ADS10292) showed more than 50% *ee*AChE inhibition, after prescreening at a concentration of 10 µM (IC_50_ = 10.88 µM). Eleven compounds were qualified for determining the IC_50_ value against *eq*BuChE. The highest BuChE inhibition was observed for compounds based on Formula **D** (1.6–5.9 µM). Additionally, compounds belonging to this group showed nanomolar potency at *h*H_3_R. Compounds based on Formula **D** should be further optimized to find the appropriate balance between affinity for H_3_R and cholinesterases. It provides a good basis for structure optimization and further investigation of MTDLs useful for Alzheimer’s disease. It is worth noting that some of MDTLs described in the papers demonstrate nanomolar potency for at least one of the evaluated targets: *h*H_3_R or AChE/BuChE.

Molecular docking studies indicate the binding mode for presented compounds and reveal the essential structural elements for the interaction with H_3_R. Amino acid residues known to recognize the histamine molecule were also involved in the binding of evaluated compounds. The analyzed compounds presented very consistent binding modes in the piperidine and piperazine derivatives of disubstituted guanidines.

The metabolic stability of ADS10310 was investigated and three metabolites were identified. After incubation with HLMs, 73.6% of ADS10310 remained in the reaction mixture, compared to only 30.8% of the unstable reference drug Verapamil. Therefore, the tested compound was found to be more metabolically stable. ADS10310 did not show any influence on CYP3A4 activity, which is responsible for the metabolism of 40–50% of all marketed drugs. Nevertheless, ADS10310 was found to be a strong inhibitor of CYP2D6 activity, which increases the risk of potential DDIs when co-administered with CYP2D6-metabolized drugs. ADS10310 showed no decrease of hepatoma HepG2 cell viability at concentrations up to 50 µM. A statistically significant decrease of hepatoma cell viability (~40%) was observed at a high dose of 100 µM. In general, ADS 10310 appears to demonstrate weak hepatotoxic activity.

## Data Availability

Data is contained within the article and Appendix A.

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
