# Peer review of "Guanidines: Synthesis of Novel Histamine H3R Antagonists with Additional Breast Anticancer Activity and Cholinesterases Inhibitory Effect"

_pharmaceuticals, 2023, doi:10.3390/ph16050675_

Round 1

Reviewer 1 Report

Dear authors

The MS entitled “Guanidines: Synthesis of Novel Histamine H3R Antagonists with Additional Breast Anticancer Activity and Cholinesterases Inhibitory Effect” was thoroughly checked for its scientific soundness. The work is noteworthy and presented in a very good manner. Almost all the experiments are performed with utmost precise. In my point of view, multi targeted approach has been well analyzed. Some of my queries are:

1. Which active site in AChE was inhibited?

2. Do the authors found any muscarinic receptors stimulation?

3. Existing such amounts of compound “ADS10310” can also reflect the slow release on site or less metabolism? kindly explain.

4. the possible intoxication that could be caused by ADS10310 and excretion mechanism.

5. Make sure the schemes are simpler.

6. make the references according to journal format.

7. Is H3R 3D structure available in PDB?

The English language is alright.

Reviewer 2 Report

REVIEW COMMENTS

Journal: Pharmaceuticals

Title: Guanidines: Synthesis of Novel Histamine H3R Antagonists with Additional Breast Anticancer Activity and Cholinesterases Inhibitory Effect

The current manuscript involves the synthesis of piperazine and piperidine-containing guanidine derivatives. A library of molecules has been synthesized and characterized. These molecules were evaluated for cytotoxicity using MTT assay in two breast cancer cell lines MDA-MB-231 and MCF7, which are known to overexpress human histamine H3 receptors. Compound ADS10310 was the lead candidate which showed 127 nanomolar potency for inhibition of hH3R, induced apoptosis in breast cancer cells, and good in vitro metabolic stability and weak hepatotoxicity in HepG2 cells. However, ADS10310 treatment resulted in potent CYP2D6 inhibition. The data suggest that these derivatives could be developed further for the treatment of Alzheimer's disease. Overall, the studies performed in the manuscript are relevant, however, some minor revisions should be made.

Minor revisions:

1) Figure 2: p should be italicized.

2) Figure 3: Scale for the microscopy images should be indicated within the image in µm.

3) HepG2 culture conditions should be included in the methods section.

4) Representative IC50 graphs for cytotoxicity in MDA-MB-231, MCF7, BJ and HepG2 cells should be provided.

5) In supplementary information, all the 1H NMRs should include the integral values and ppm of proton peaks and the 13C NMRs should include the ppm values of carbon peaks in the images.
